# Effects of Fe$_2$O$_3$ on Reduction Process of Cr-Containing Solid Waste Self-Reduction Briquette and Relevant Mechanism

**Tuo Wu, Yanling Zhang \*, Zheng Zhao and Fang Yuan**

State Key Laboratory of Advanced Metallurgy, University of Science and Technology Beijing, Beijing 100083, China; wutuo90@163.com (T.W.); ext_zheng@163.com (Z.Z.); 408995357@163.com (F.Y.)

\* Correspondence: zhangyanling@metall.ustb.edu.cn; Tel.: +86-139-1189-1432

**Abstract:** High-temperature quench method, scanning electron microscope-energy dispersive spectroscopy (SEM-EDS), and thermodynamic analysis were adopted to study the effects of Fe$_2$O$_3$ on reduction process of Cr-containing solid waste self-reduction briquette (Cr-RB). Moreover, the relevant mechanism was also studied. The results clearly showed that the addition of Fe$_2$O$_3$ decreased the chromium-iron ratio (Cr/(Fe + Cr)) of Cr-RB itself and promoted the reduction of chrome oxide in the Cr-containing solid wastes such as stainless steel slag and dust. A large number of Fe-C alloy droplets generated in the lower temperature could decrease the activity of reduced chromium by in situ dissolution and the reduction of Cr-oxide was accelerated. Rapid separation of metal and slag could be achieved at a relatively lower temperature, which was very beneficial to the efficient recovery of Cr. Finally, the corresponding mechanism diagram was presented.

**Keywords:** Cr recovery; self-reduction briquette; reaction mechanism

---

## 1. Introduction

In the entire process of stainless steel smelting, chromium is inevitably oxidized into stainless steel slag (SSS) [1,2], especially for the EAF (electric-arc furnace) smelting process, stainless steel dust (SSD) [3,4], and mill scales [5] to form Cr-containing solid wastes. These solid wastes are valuable secondary resources, especially containing a considerable amount of chromium metal. Fully recycling chromium in these solid wastes will not only improve enterprises' economic efficiency, but also reduce the risk of heavy metal pollution caused by Cr in solid wastes after being leached in the natural acidic or alkaline environment [6]. In all of Cr-containing solid wastes treatment technologies, high-temperature reduction technology using carbonaceous reductants has been paid extensive attention due to its unique characteristics compatible with metallurgical enterprises [5,7–10]. Due to its excellent reactivity, self-reduction briquette bearing C was mostly adopted [11–14]. The composition design of self-reduction briquette is critical to efficient recovery of Cr as the high temperature reduction technology is applied. Studies have shown that FeO$_x$ ($x$ = 0, 1, 4/3) can significantly affect the carbothermal reduction process of chromium oxides which mainly was the bulk phase in the Cr-containing solid wastes. Görnerup et al. [15,16] studied the reduction behavior of FeO and Cr$_2$O$_3$ in stainless steel slag and found that the reduction of Cr$_2$O$_3$ by C had a pronounced incubation period and the reduction of FeO did not have any incubation period. The reduction of Cr$_2$O$_3$ did not start until some liquid Fe-C alloys had formed and the product was Fe-Cr-C alloy. Abundant FeO in the slag was beneficial to rapid reduction of Cr$_2$O$_3$. Chakraborty et al. [17] investigated the carbothermal reduction of chrome ore ((Fe, Mg)O·(Fe, Cr, Al)$_2$O$_3$) and found that iron had been fully reduced before the reduction of Cr. Hu et al. [18–20] studied the alloying effect of special precursor made up of

chromite ore, mill scale, and carbon, and pointed out that the Cr/(Cr + Fe) of the alloy precursor had important influence on the yield of Cr. Therefore, the introduction of $FeO_x$ into Cr-containing solid waste self-reduction briquette (Cr-RB, for short) is very necessary. However, the intrinsic behaviour of reduction and melting-separation processes of Cr-RB with the addition of $FeO_x$ has not been fully investigated. Accordingly, high temperature reduction-rapid quench, SEM-EDS, and thermodynamic analysis methods were adopted to clarify this problem for the Cr-RB with the addition of $Fe_2O_3$ or not (A: $Fe_2O_3$ + SSS + SSD + C, B: SSS + SSD + C). A detailed analysis of reduction results at different temperatures and times would contribute to understand the relevant mechanism of reduction process of Cr-containing solid waste self-reduction briquette.

## 2. Materials and Methods

### 2.1. Materials

Figure 1 shows the three experimental materials, which are SSD (AOD dust), SSS (EAF slag), and $Fe_2O_3$ respectively. The SSS was coarsely crushed by a jaw crusher and finely ground by an ore crusher. Then, metallic iron particles contained in it were removed by a magnet. The powdery slag and the SSD were all passed through a 60-mesh steel screen in order to eliminate the effect caused by the uneven particle size. Reductant used in the experiment was high-purity graphite powder (C $\geq$99.85%) and the flux was analytical reagent $SiO_2$. All materials were dried at 200 °C for 24 h before experiment. The composition of three experimental materials was shown in Table 1.

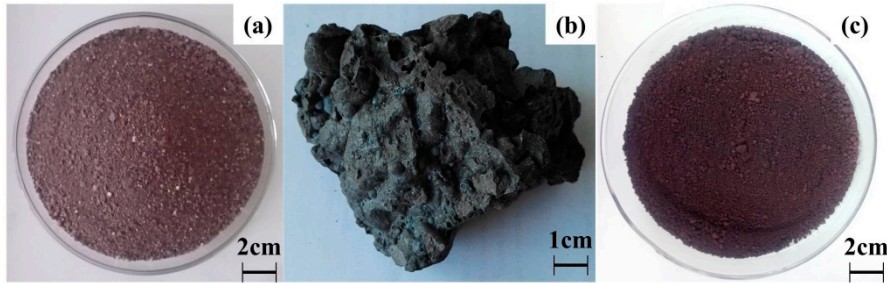

**Figure 1.** Experimental material (**a**) stainless steel dust (SSD) (**b**) stainless steel slag (SSS) (**c**) $Fe_2O_3$.

**Table 1.** Composition of experimental materials wt.%.

| Mater. | TFe | Cr | CaO | SiO$_2$ | MgO | Al$_2$O$_3$ | NiO | MnO | MoO$_3$ | V$_2$O$_5$ | F | Others |
|---|---|---|---|---|---|---|---|---|---|---|---|---|
| SSD | 20.18 | 35.80 | 20.67 | 3.10 | 3.49 | 1.70 | 0.11 | | 0.54 | 0.10 | | Bal. |
| SSS | 3.20 | 9.12 | 41.01 | 28.22 | 9.43 | 2.14 | | 1.50 | | 0.28 | 0.64 | Bal. |
| Fe$_2$O$_3$ | $\geq$99% | | | | | | | | | | | Bal. |

Figure 2 shows the XRD spectrum of SSD and SSS. It can be seen that the SSD mainly contains CaO, $FeCr_2O_4$, $Fe_3O_4$, $MgFe_2O_4$, and small amounts of $Ca_3Mg(SiO_4)_2$ and metal Fe. Among them, the valuable elements Fe and Cr mainly existed in the spinel phase. Due to the isomorphism phenomenon of spinel phase, Fe and Cr in the dust actually were in a solid solution similar to $(Fe, Mg)O \cdot (Fe, Cr)_2O_3$. The SSS mainly contained $MgCr_2O_4$, $Ca_3Mg(SiO_4)_2$ and Fe, in which Cr was mainly in the Magnesiochromite ($MgCr_2O_4$) with a high melting point (2390 °C). In summary, most of the valuable metals Fe and Cr in the Cr-containing solid wastes were in their corresponding spinel solid solution phase.

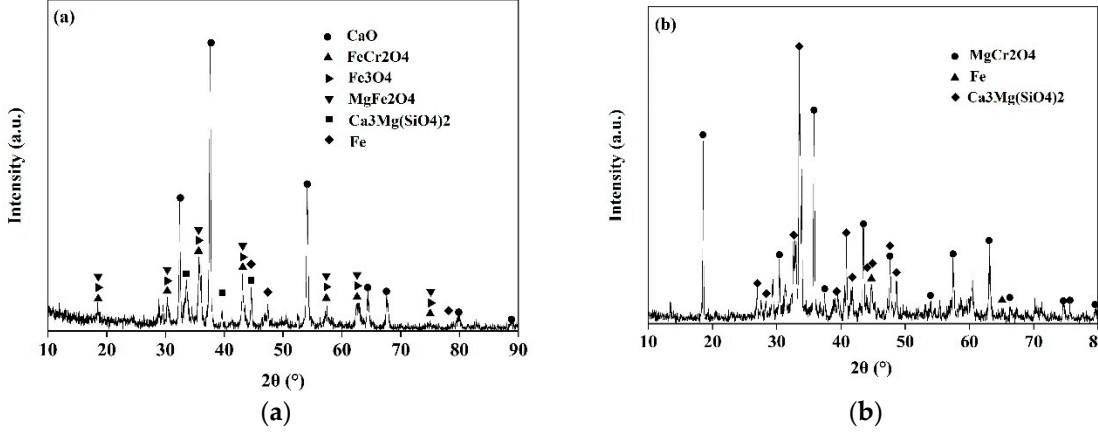

**Figure 2.** XRD spectrum of SSD (**a**) and SSS (**b**).

## 2.2. Apparatus and Procedure

According to the proportion scheme shown in Table 2, the experimental materials were accurately weighed and thoroughly mixed in a mixer. Then, for A and B groups, nine parts of mixture with a quantity of about 10 g were weighed. The powder mixture was made into a self-reduction briquette by a constant pressure of 30 MPa for 2 min in a steel mold (Inner diameter: 20 mm), and then put in a high-purity MgO crucible (Outside diameter: 28 mm, Inner diameter: 21 mm, Height: 68 mm). A vertical tube furnace (Version: BLMT-1700 °C) heated by six U-shaped MoSi$_2$ heating elements was used for the reduction experiment. The schematic of the experimental set-up was shown in Figure 3. Its constant temperature zone with fluctuation of ±2 °C was 6 cm, and the temperature control accuracy was ±1 °C.

**Table 2.** Experiment scheme.

| Proportion Scheme | Temperature | Holding Time/Min | | | Cr/(Cr + Fe) |
|---|---|---|---|---|---|
| A | L: 1350 | 5 | 20 | 40 | |
| (30%Dust+20%SSS+50%Fe$_2$O$_3$): 100 g Graphite: $n_C$:$n_O$ = 1.2 SiO$_2$: R = 1.2 | M: 1450 | 5 | 20 | 40 | 0.23 |
| | H: 1550 | 5 | 20 | 40 | |
| B | L: 1350 | 5 | 20 | 40 | |
| (60%Dust+40%SSS): 100 g Graphite: $n_C$:$n_O$ = 1.2 SiO$_2$: R = 1.2 | M: 1450 | 5 | 20 | 40 | 0.65 |
| | H: 1550 | 5 | 20 | 40 | |

$n_C$ is the number of C moles for graphite, and $n_O$ is the number of O moles in FeO$_x$ and Cr$_2$O$_3$ from slag, dust and Fe$_2$O$_3$. R is the basicity of briquette, (wt.% CaO)/(wt.% SiO$_2$).

First, high-purity Ar (1 L/min, ≥99.999%) was introduced into the furnace tube throughout the entire experiment before the temperature control program was run. When the temperature was raised to the target (1350 °C, 1450 °C, 1550 °C) and stabilized, the crucible containing the sample was preheated in the upper low temperature zone for 1.5 min and then placed in the constant temperature zone for 5, 20, 40 min as shown in Table 2. Once the holding time was finished, the hot sample was rapidly taken out from furnace tube and quenched by the high-purity Ar. The quench time was about 2 min. When the crucible cooled down to room temperature, its weight loss ratio was calculated. Finally, the crucible containing the reaction product was integrally inlaid by resin and longitudinally cut. One half of it was ground, polished, and sprayed with gold to meet demand of SEM. For convenience, each sample was numbered. For example, AM20 represented a sample of A which was heated at 1450 °C for 20 min.

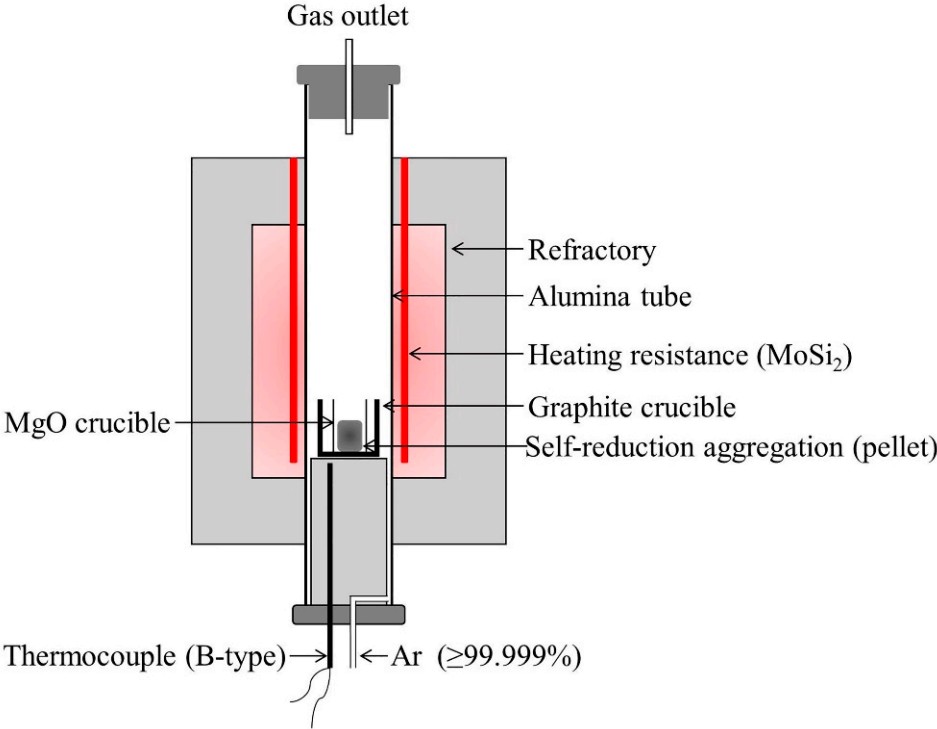

**Figure 3.** Schematic of tube furnace.

## 3. Results and Discussion

### 3.1. Reduction Yield

Based on the mass variation of reaction system before and after high temperature reduction, the corresponding weight loss could be obtained, and the reduction yield ($\varphi$) could be calculated according to Equation (1), where $M_{(weight\ loss)}$ is actual mass loss and $M_{(theory\ loss)}$ is theoretical mass loss calculated as the maximum mass loss due to assumed emission of CO produced after the complete reduction of $Cr_2O_3$ and $FeO_x$ in the self-reduction briquette. Since reduction experiments were carried out at relatively high temperatures ($\geq 1350\ °C$), in which CO was more stable than $CO_2$ and high-purity argon would take the gas product continuously, and an excessive amount of C could react with $CO_2$ produced by indirect reduction (Equation (2)) to form CO based on Boudouard reaction (Equation (3)), it was assumed that the gas product is mainly CO (Equation (4)) when the Equations (2) and (3) were combined.

$$\varphi = \frac{M_{(weight\ loss)}}{M_{(theory\ loss)}} \times 100\% \tag{1}$$

$$Me_xO_y(s) + CO(g) = Me(s) + y/2CO_2 \text{ (Me: Fe and Cr)} \tag{2}$$

$$y/2CO_2(g) + yC(s) = 2yCO(g) \tag{3}$$

$$yC(s) + Me_xO_y(s) = xMe(s) + yCO(g) \text{ (Me: Fe and Cr)} \tag{4}$$

Figure 4 shows the time and temperature dependences of reduction yield of A and B-type self-reduction briquettes. As shown that the reduction yield of them increases rapidly before 5 min and then its increasing rate slows down with the extension of time. In addition, higher temperature results with higher reduction yield for the A-type Cr-RB after 5 min, which reflects the superiority of carbothermal reduction reaction at higher temperature. However, the reduction yield of the B-type Cr-RB does not show the same trend after 5 min, which may be due to the samples' easier reoxidation caused by quench operation. Compared with the low-Cr metal product ($Cr/(Cr + Fe) = 0.23$) in the A-type self-reduction briquette, high-Cr metal product ($Cr/(Cr + Fe) = 0.65$) in the B-type self-reduction

briquette has higher Cr activity coefficient especially at higher temperature, which leads to the increase of $\Delta G$ of Reaction (4) and the decrease of reduction yield. In general, the reduction yield of the A-type Cr-RB is higher than that of the B-type self-reduction briquette, which reflects that $Fe_2O_3$ can significantly improve the reducibility of B-type Cr-RB.

$$(Fe, Cr)(s) + C(s) = [Fe, Cr, C](l) \tag{5}$$

$$y[C] + Me_xO_y(s, l) = xMe(s) + yCO(g) \text{ (Me: Fe and Cr)} \tag{6}$$

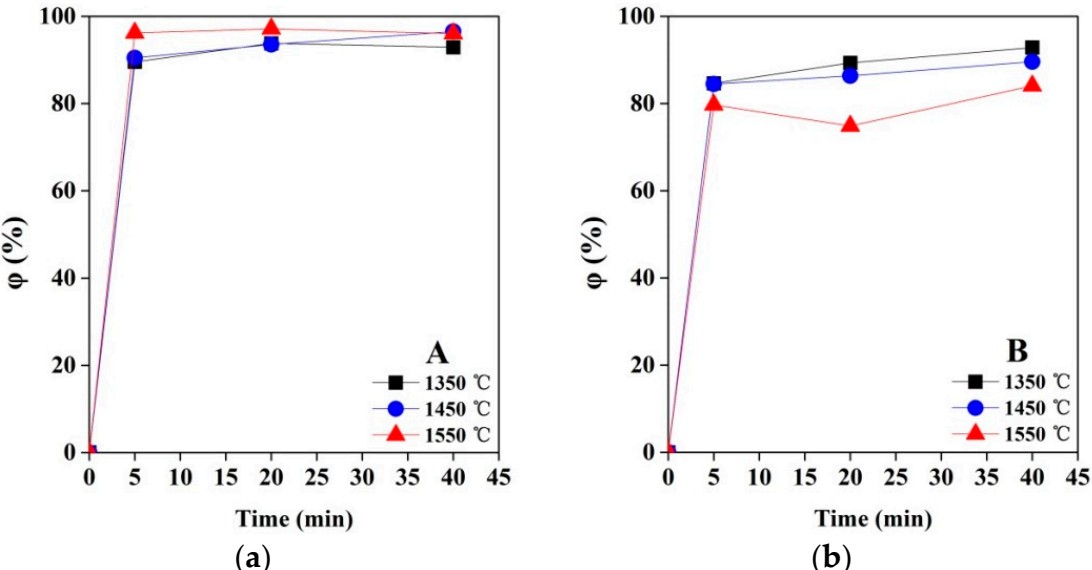

**Figure 4.** Time and temperature dependences of reduction yield of A (**a**) and B-type (**b**) self-reduction briquettes.

Obviously, the self-reduction reaction of briquettes was basically finished within 5 min according to Reaction (4). The carburization reaction (Reaction (5)) also occurred subsequently. After that, the dissolved [C] would react with the Fe or Cr oxides by Reaction (6). However, due to the decrease in activities of C and oxides, the reduction yield increased slowly after 5 min.

*3.2. Morphology of Products*

Figure 5 shows the longitudinal section overview of the reduction products at different temperatures and times. It can be seen that as the temperature keeps constant, the metal products in every samples gradually contact each other and grow up, and the metal-slag interface are clearer which means the separation of them are more complete as the reaction time prolongs. The same phenomenon also occurred for the increasing temperature when reaction time remained constant. Compared with the sample BL5, sample AL5 has better reactivity, and many small metal particles are clearly visible in the reduction products after holding at 1350 °C for 5 min. Although BL5 has a violent reaction seen from many holes caused by gas product, no clear metal particles are formed, and slag and metal mixes together. As seen in AM20 sample, a good separation between slag and metal is achieved after holding at 1450 °C for 20 min. But for the corresponding BM20 sample, a holding of 20 min at 1550 °C must be needed. It is observed that the addition of $Fe_2O_3$ was very effective for rapid slag-metal separation from the Cr-RB at a relatively low temperature.

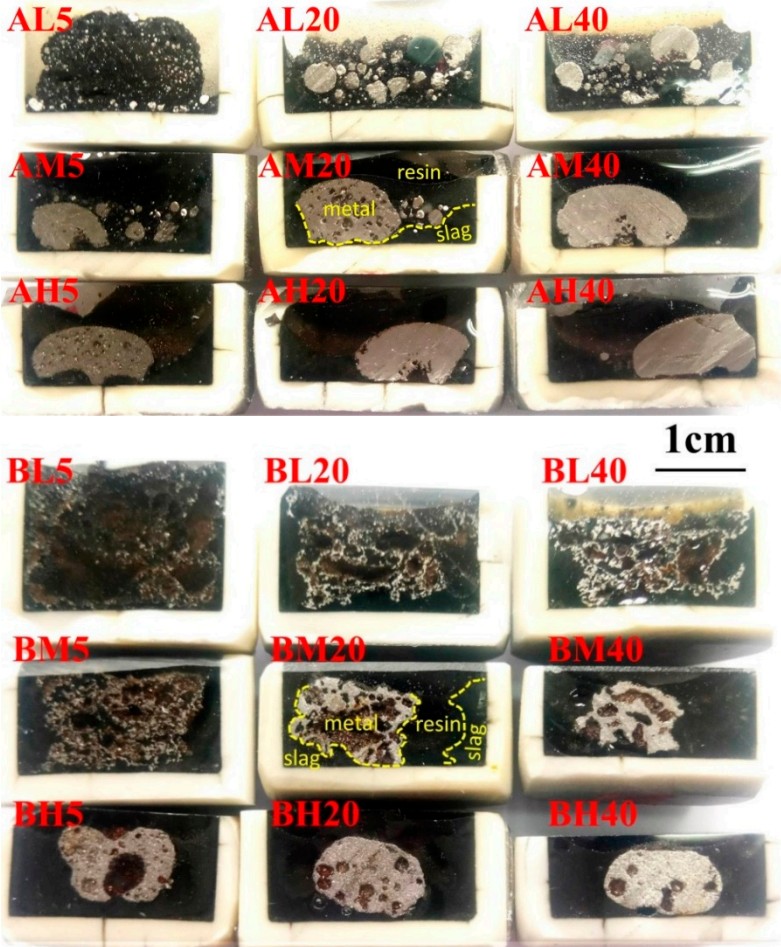

**Figure 5.** Longitudinal section overview of the reduction product.

### 3.3. Variation of Product Morphology Observed by SEM

Figures 6 and 7 are the morphologies of the reaction products of Cr-RB under different conditions observed by SEM. It can be further seen that the increase of reaction temperature and the prolongation of reaction time facilitate the effective separation of the metal product from the slag. Higher temperature increases carbothermal reduction rate of Fe or Cr oxides [21,22]. In addition, the increase in temperature enhances the fluidity of metal and slag, and accelerates the separation between them, which make metal aggregate, grow, and deposit more easily. The prolongation of reaction time and the increase of temperature are in favor of metal recovery from Cr-containing solid wastes, especially for the Cr-RB with some $Fe_2O_3$. Moreover, the metal produced by A is mostly spherical before the metal particles completely aggregates, and its contact surface with the slag is small to separate; the metals in B are mostly sharp-edged blocks or strips, and they are mutually overlapped with slag and is not easy to separate. The above results indicate that the rapid reduction of chromium oxides in Cr-containing solid waste can be achieved while accompanying the reduction of iron oxides.

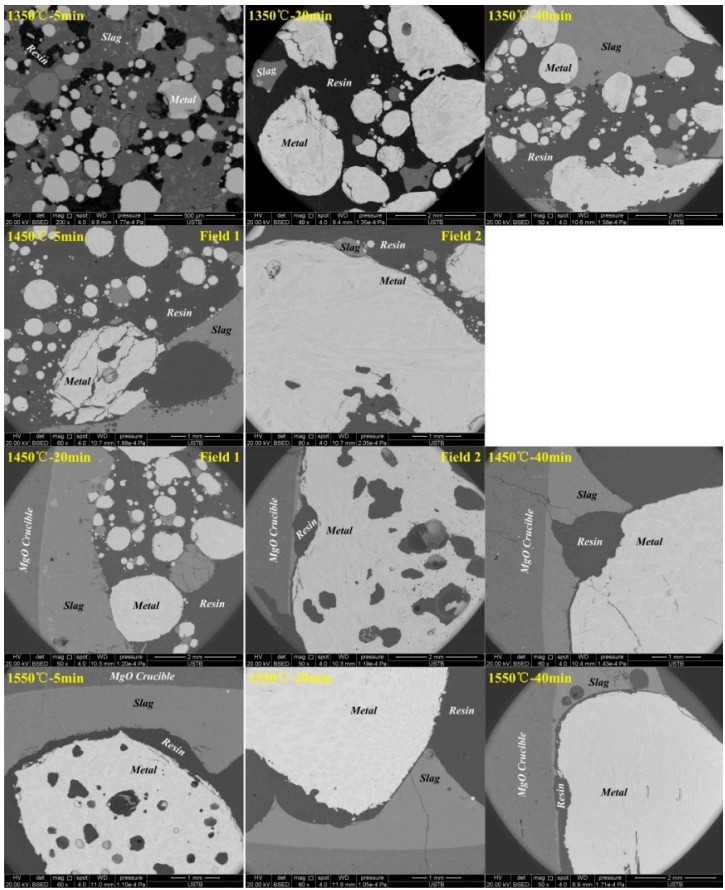

**Figure 6.** Morphology of the reduction product of A-type Cr-containing solid waste self-reduction briquette (Cr-RB) under different conditions observed by SEM.

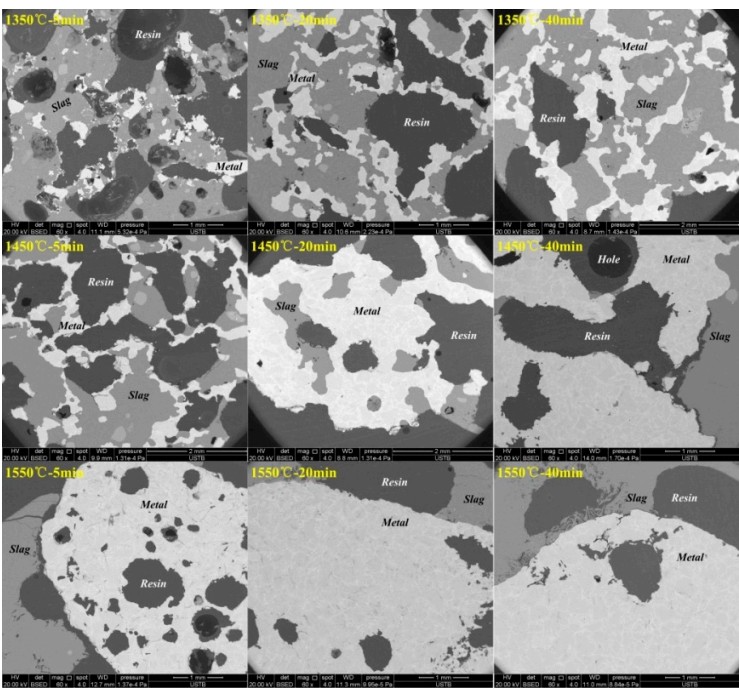

**Figure 7.** Morphology of the reduction product of B-type Cr-RB under different conditions observed by SEM.

Figures 8 and 9 show the morphology, element distribution, and EDS spectra of typical phase in AL5 and BL5 samples. It can be found that a large number of Fe-Cr-C metal particles were formed in AL5 (Figure 8). When the Cr concentration is low, it is mostly spherical (point 1). When the Cr concentration is high, the shape is irregular (point 9). There are also (Fe, Mg)O·(Cr, Al)$_2$O$_3$ spinels (point 13), which are from the raw material, not yet sufficiently reduced, and a slag phase (point 11) where the Cr concentration reaches 3.15%. Similar to AL5, a large number of irregular Cr-Fe-C alloys with high Cr concentration (point 1), slag (point 11), and insufficient reduction of (Fe, Mg)O·(Cr, Al)$_2$O$_3$ spinel (point 13) were generated in BL5 (Figure 9). The element distribution results showed that in the reduction process, a large number of spherical Fe-based metal droplets were generated in AL5, and a certain concentration of Cr and C dissolved in them; a large amount of Cr-based metals were produced in BL5, and a certain concentration of Fe and C dissolved in them too. This further reflects that the relative proportion of Fe and Cr in the Cr-RB would affect the speed of the entire reduction progress and the recovery of Cr element. Studies [23] have pointed out that chromium ore with higher iron oxide content can achieve higher reduction at relatively low temperatures. In the study of the carbothermal reduction process of the stainless steel slag in the electric furnace, Görnerup et al. [15,16] obtained that the increase of the liquid metal containing C in the slag will promote the recovery of Cr and suggested to appropriately increase the FeO content in the slag. Hu et al. [20] pointed out that Cr/(Fe + Cr) was a critical parameter affecting the recovery of Cr in alloying agents when studying the direct alloying of steel by chromium ore, and thought that the alloying precursor with low Cr/(Fe + Cr) were more likely to obtain high Cr recovery. In this study, the Cr/(Fe + Cr) of A and B-Type self-reduction aggregation are 0.23 and 0.65, respectively. So the former is more efficient for the recovery of Cr.

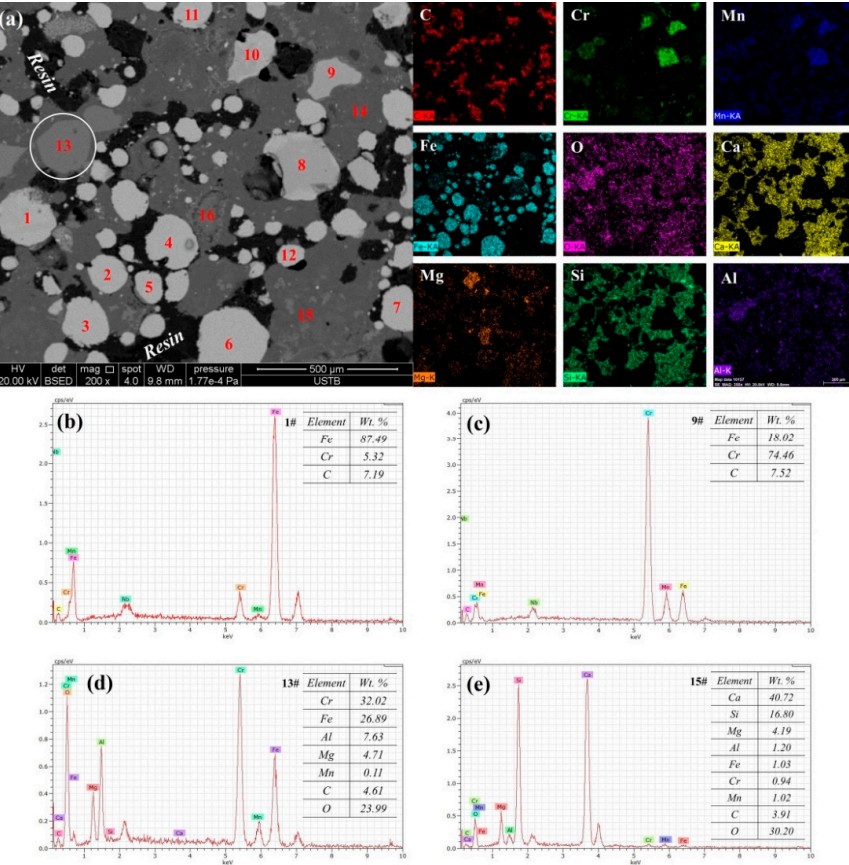

**Figure 8.** Element distribution of AL5 sample (**a**), EDS spectra of typical phases Point 1 (**b**), Point 9 (**c**), Point 13 (**d**) and Point 15 (**e**).

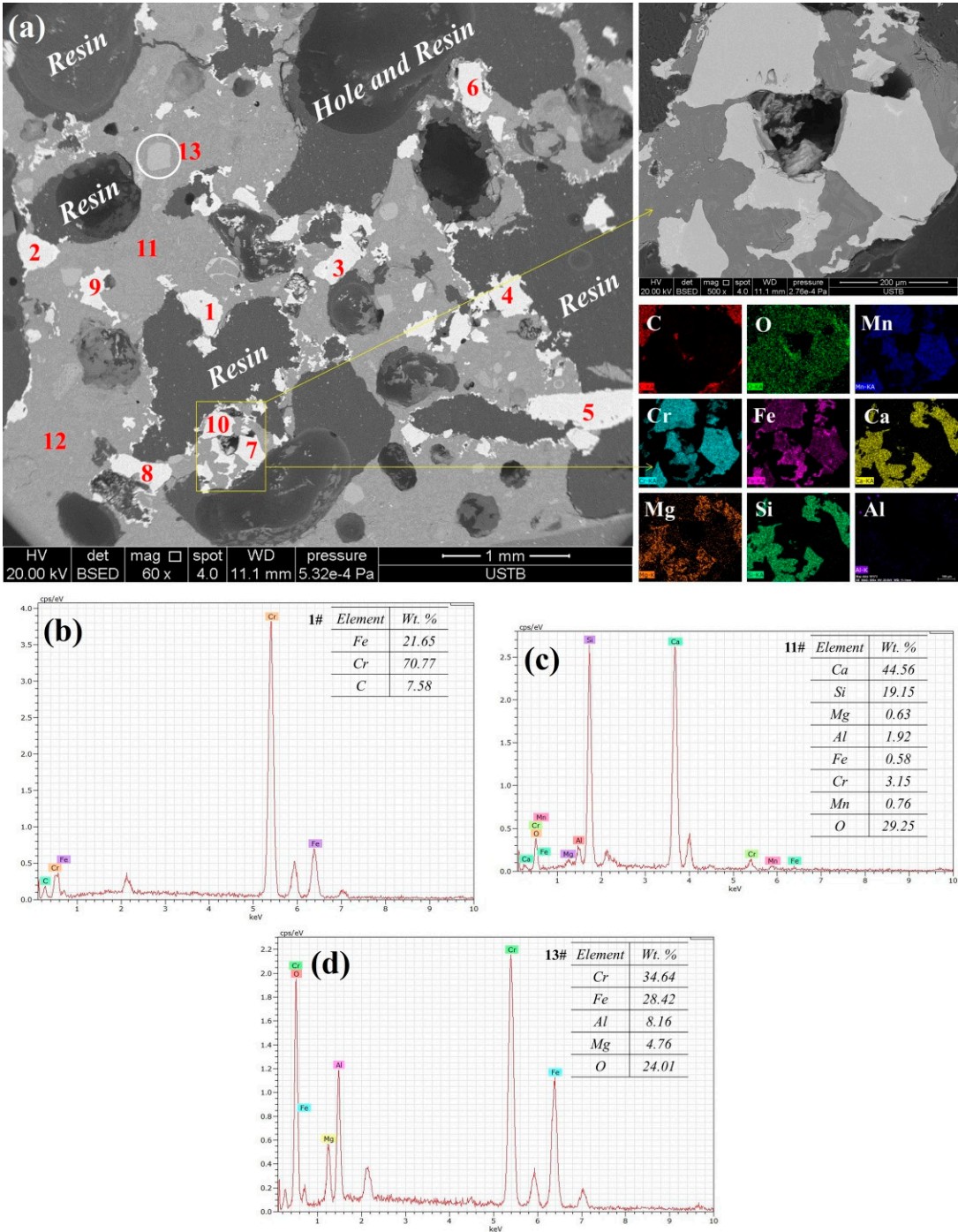

**Figure 9.** Element distribution of BL5 sample (**a**), EDS spectra of typical phases Point 1 (**b**), Point 11 (**c**), and Point 13 (**d**).

Further comparison of the two slags after holding at 1450 °C for 5 min was made, and the results revealed that there were residual Cr-containing spinel phases (($Fe$, $Mg$)$O\cdot$($Cr$, $Al$)$_2O_3$) remaining in the BM5 sample as shown in Figure 10, which came from SSS or SSD. However, this phase was rarely seen in the slag of the AM5 sample. This indicated that the Cr-containing phase could not be rapidly reduced in the B-type Cr-RB without $Fe_2O_3$, which corresponded to that the reduction yield of the A-type Cr-RB was higher than that of the B-type Cr-RB at 5 min.

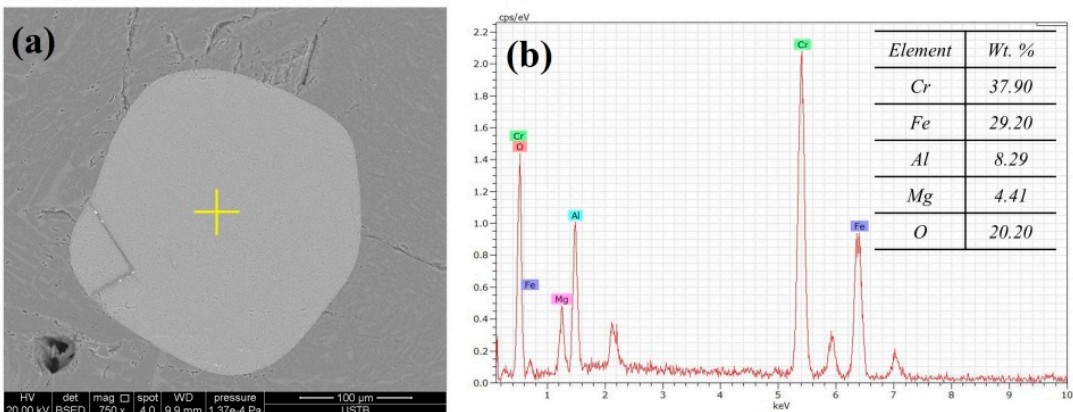

**Figure 10.** Residual (Fe, Mg)O·(Cr, Al)$_2$O$_3$ Spinel in BM5 sample, (**a**) morphology and (**b**) EDS.

### 3.4. Variation of Metal Composition

Base on EDS analysis, the composition of different small metal particles and large metal product in the observation field of SEM could be obtained. By means of Fe-Cr-C ternary phase diagram, the variation of metal composition in the samples at different conditions could be clearly observed, which was shown in Figure 11. As shown that the liquidus temperature increased with increasing concentration of C and Cr. For example, the liquidus temperature increased to 1575 °C from 1314 °C when the Cr concentration increased to 70 wt.% from 10 wt.% for the Fe-6 wt.%C-Cr system. Specifically for the AL5 sample, the metal products mostly distributed near the liquidus of 1350 °C, and C in the metals was in a saturated state. A few metal products were in the M7C3 ((Cr, Fe)$_7$C$_3$) and liquid phase coexistence zone, which was away from the liquidus. The Cr concentration in the metals increased with increasing reaction time, which indicated that the metal with a high Cr concentration and the metal with a low Cr concentration contacted and dissolved each other. When the temperature was 1450 °C and 1550 °C, the bulk metal component was in the liquid phase zone and had good fluidity and reactivity. For BL5 sample, the metal composition was far away from the liquid zone, and its melting point was as high as 1650–1750 °C. The metals contained a large amount of high-melting phase M7C3, which increased the viscosity of the metal [24] and was not conducive to the separation of metal and slag. The increase of the temperature to 1450 °C and 1550 °C accelerated the metal agglomeration, but its composition was still outside the liquid zone of the corresponding temperature, and M7C3 still existed.

Figure 12 is a Fe-Cr-C ternary phase diagram with iso-activity line of Cr at 1450 °C, in which the metal product composition of A and B-type Cr-RB at different temperature and holding time were projected. As shown that the Cr activity depended intensely on the Cr concentration once the C concentration was relatively constant. For example, the Cr activity increased to 0.5 from 0.02 when the Cr concentration increased to 70 wt.% from 10 wt.% for the Fe-6 wt.%C-Cr system. Obviously, the Cr activity of the metals in the samples of group A was significantly lower than that of the metals in the samples of group B. Higher Cr activity in the metals was not conducive to Cr transmit from slag into metal. Thus, the Cr in the A-type Cr-RB tended to move toward the metal faster.

In conclusion, due to the relatively low chromium-iron ratio (Cr/(Fe + Cr)) of the A-type Cr-RB, the metal products are more likely to rapidly aggregate and separate from the slag. The low Cr concentration of the metal products in the A-type Cr-RB is more thermodynamically beneficial for Cr recovery.

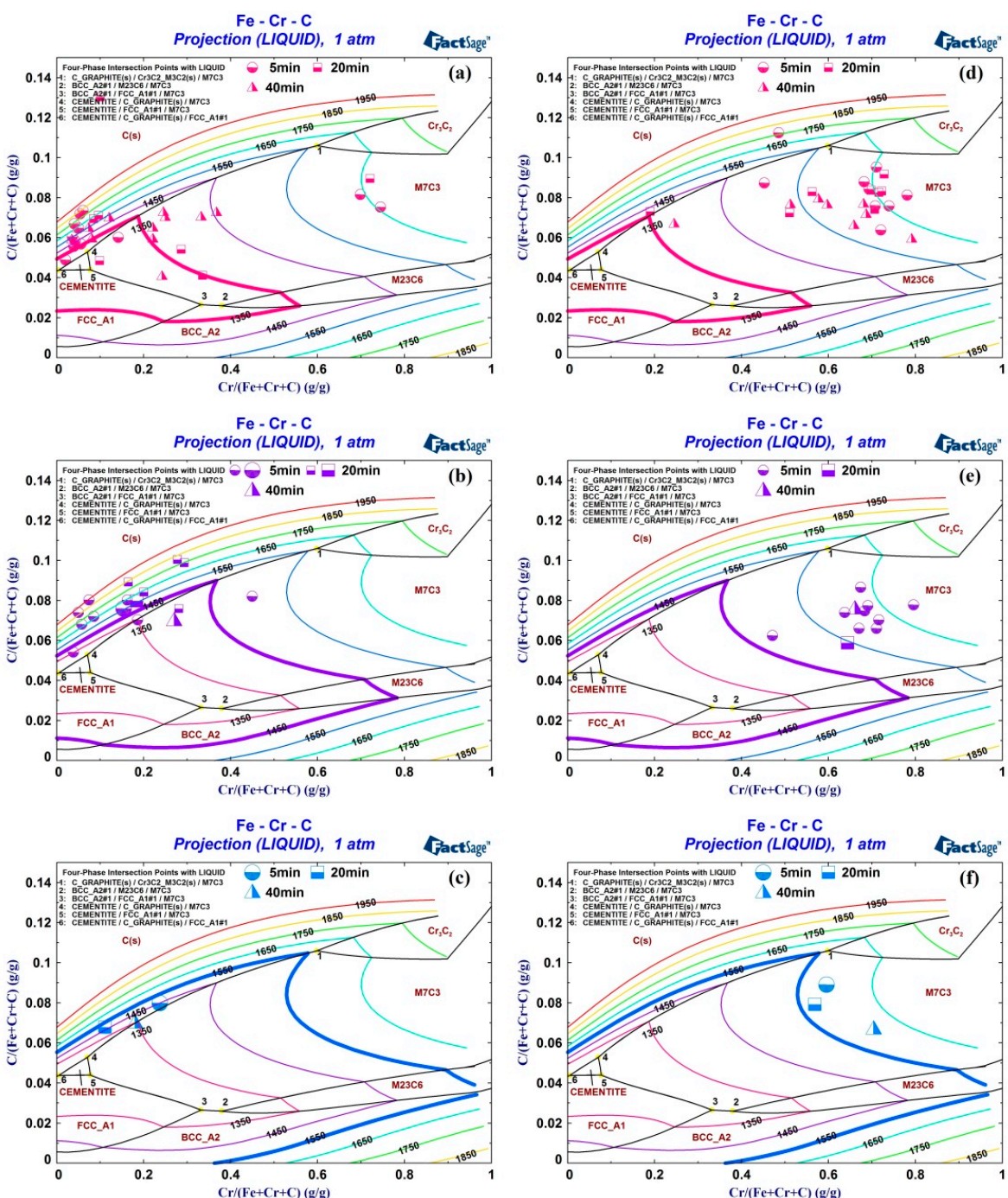

**Figure 11.** Metal composition variation in A and B-type Cr-RBs at different temperatures and times, (**a**) A-1350 °C, (**b**) A-1450 °C, (**c**) A-1550 °C, (**d**) B-1350 °C, (**e**) B-1450 °C, (**f**) B-1550 °C (Calculated by FactSage7.0, FSstel Database).

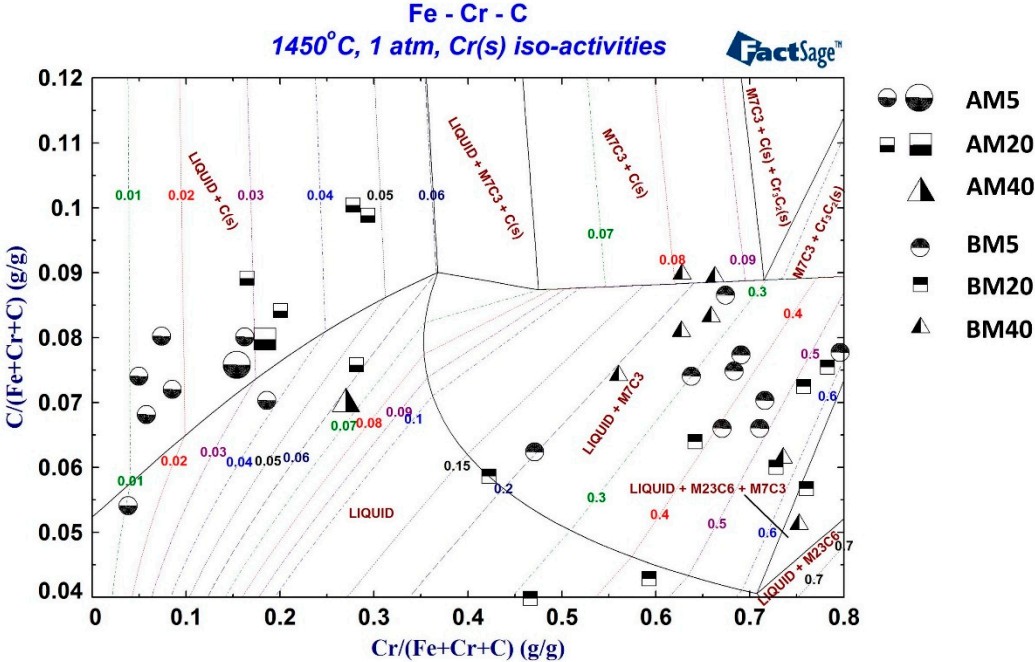

**Figure 12.** Activity of Cr in metals at 1450 °C (pure Cr as standard state, calculated by FactSage 7.0, FSstel Database).

### 3.5. Reaction Mechanism

Figure 13 shows that the initial reaction temperature ($T_{initial}$) of Reactions (7) to (14) in the standard state are 712 °C, 655 °C, 653 °C, 1053 °C, 1248 °C, 1141 °C, 1253 °C, 1710 °C, respectively. Obviously, the initial reduction temperatures of iron-containing oxides are below 712 °C. Once the Fe was first reduced, it could be carburized by C or CO to produce a large amount of liquid Fe-C alloys (Reaction (15)) [25,26]. The initial reduction temperatures of the Cr-containing oxides are all above 1000 °C, and especially for the reaction between $Cr_2O_3$ and $Cr_7C_3$, the $T_{initial}$ reaches to 1710 °C. Research [27] has pointed out that the carbothermal reduction of $Cr_2O_3$ was carried out according to $Cr_2O_3 \rightarrow Cr_3C_2 \rightarrow Cr_7C_3 \rightarrow Cr_{23}C_6 \rightarrow Cr$, and the stability range of different chromium carbides was different [23] ($Cr_7C_3$: 1250–1600 °C). Therefore, $Cr_7C_3$ produced at the experimental temperature would delay the transition of $Cr_2O_3$ to Cr. When the Cr-RB had a lower Cr/(Fe + Cr), a large amount of Fe-C alloy droplets would be generated before the reduction of $Cr_2O_3$. The pure Cr or $Cr_7C_3$ produced later would in-situ dissolve into the Fe-C alloys, and the liquid Fe-Cr-C alloy was formed (Reaction (16)). Meanwhile, the activity of the reduced product Cr or $Cr_7C_3$ decreased. As shown in Equations (17) and (18), the decrease in Cr activity was favorable for the decrease in $\Delta G$ of the reactions, which accelerated Cr recovery. Therefore, the reduction efficiency of Cr in A-type Cr-RB with $Fe_2O_3$ was higher than that of B-type Cr-RB. Based on the above experimental results and analysis, a schematic diagram of reaction mechanism was shown in Figure 14.

$$1/4Fe_3O_4(s) + C(s) = 3/4Fe(s) + CO(g) \tag{7}$$

$$1/3MgFe_2O_4(s) + C(s) = 1/3MgO(s) + 2/3Fe(s) + CO(g) \tag{8}$$

$$1/3Fe_2O_3(s) + C(s) = 2/3Fe(s) + CO(g) \tag{9}$$

$$FeCr_2O_4(s) + C(s) = Cr_2O_3(s) + Fe(s) + CO(g) \tag{10}$$

$$1/3MgCr_2O_4(s) + 9/7C(s) = 1/3MgO(s) + 2/21Cr_7C_3(s) + CO(g) \tag{11}$$

$$1/3Cr_2O_3(s) + 9/7C(s) = 2/21Cr_7C_3(s) + CO(g) \tag{12}$$

$$1/3Cr_2O_3(s) + C(s) = 2/3Cr(s) + CO(g) \tag{13}$$

$$1/3Cr_2O_3(s) + 1/3Cr_7C_3(s) = 3Cr(s) + CO(g) \tag{14}$$

$$Fe(s) + C(s) = (Fe-C)_{alloy} \tag{15}$$

$$Fe-C_{alloy} + Cr + Cr_7C_3 = (Fe-Cr-C)_{alloy} \tag{16}$$

$$\Delta G_{13} = \Delta G_{13}^0 + RT \ln \frac{\frac{P_{CO}}{P^0} \cdot a_{Cr}^{2/3}}{a_C \cdot a_{Cr_2O_3}^{1/3}} \tag{17}$$

$$\Delta G_{14} = \Delta G_{14}^0 + RT \ln \frac{\frac{P_{CO}}{P^0} \cdot a_{Cr}^3}{a_{Cr_7C_3}^{1/3} \cdot a_{Cr_2O_3}^{1/3}} \tag{18}$$

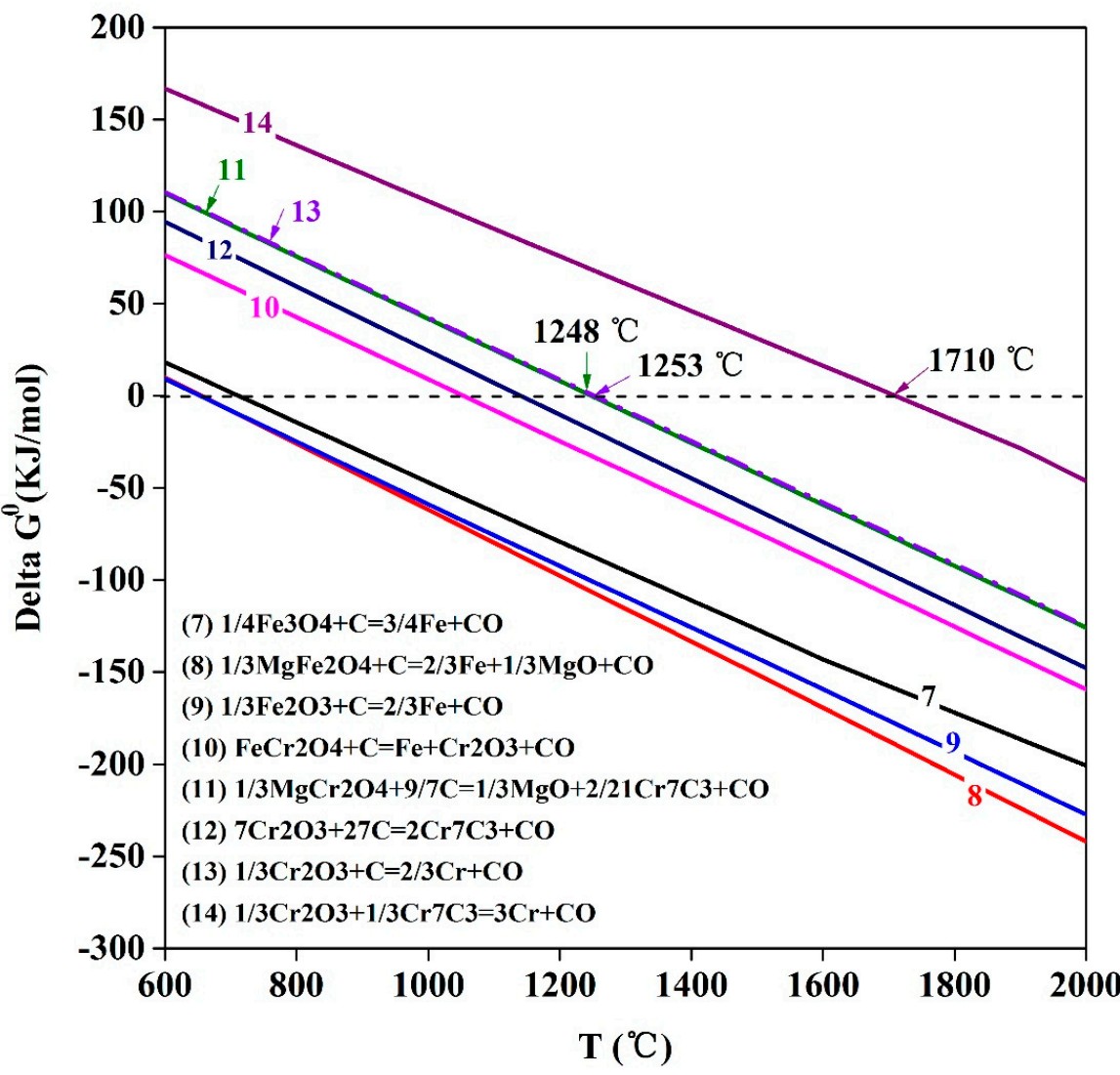

**Figure 13.** Variation of standard Gibbs free energy with temperature for each reaction (Calculated by FactSage 7.0, FactPS Database).

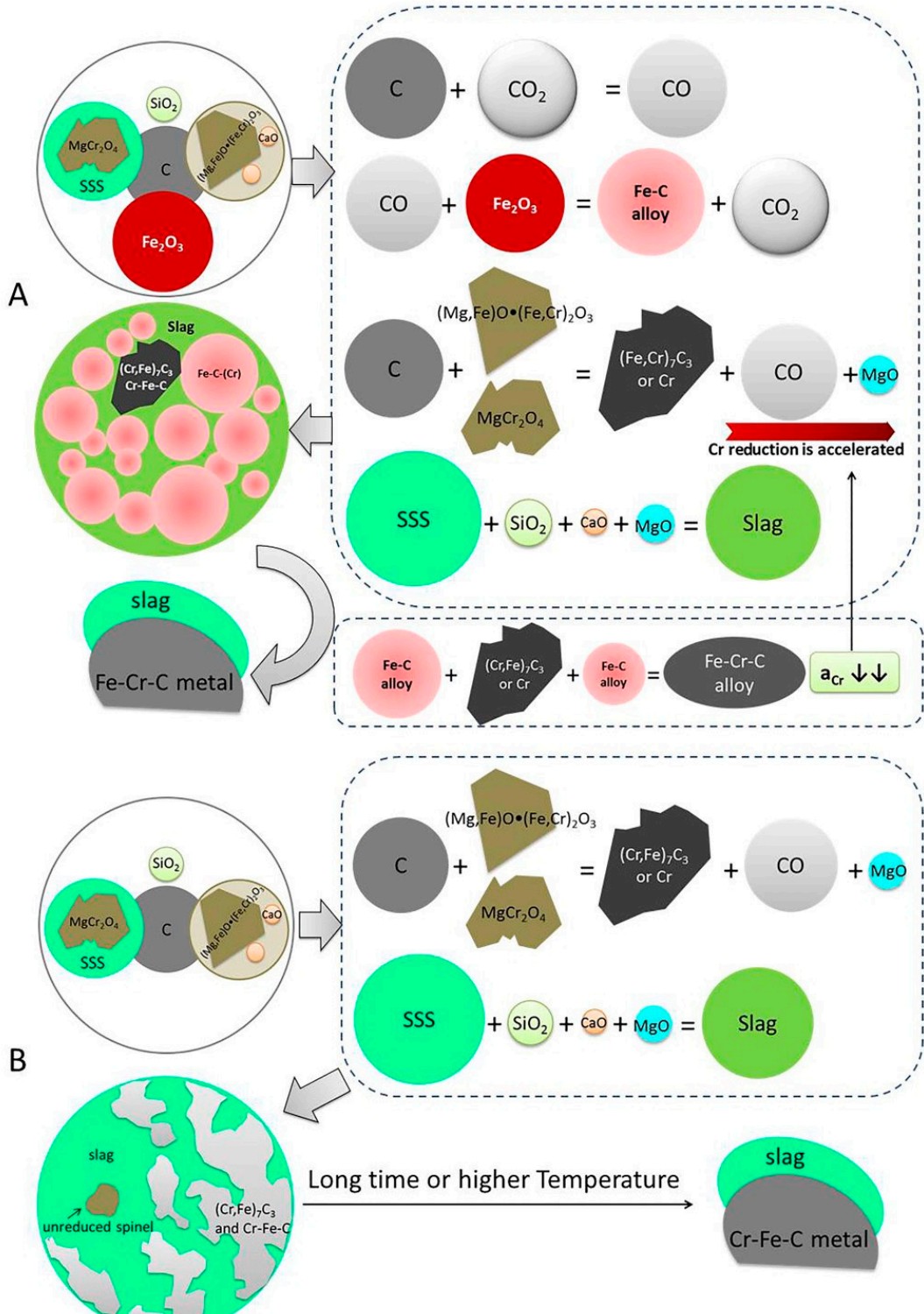

**Figure 14.** Mechanism diagram of the reduction-melting process of two Cr-RBs.

## 4. Conclusions

(1) The reduction yield of the two self-reduction briquettes A and B increased rapidly within 5 min, and then the growth rate became slower. The introduction of $Fe_2O_3$ to the Cr-containing solid waste self-reduction briquettes increased its reduction yield at a certain temperature.

(2) The addition of $Fe_2O_3$ to the Cr-containing solid waste self-reduction briquettes decreased its $Cr/(Cr + Fe)$, and a large amount of spherical Fe-C alloys with low Cr concentration was produced at a relatively low temperature. They could in-situ dilute the pure Cr or $Cr_7C_3$ to decrease the activity of Cr and promote the rapid reduction of $Cr_2O_3$, improving the recovery efficiency of Cr. Therefore, when the composition of Cr-containing solid waste self-reducing aggregation was designed for the furnaces with strong reducing atmosphere such as rotary hearth furnace, shaft furnace, or EAF furnace et al., much attention should be paid for its chromium-iron ratio $(Cr/(Fe + Cr))$.

**Author Contributions:** conceptualization, Y.Z.; methodology, T.W.; software, T.W.; validation, Y.Z.; formal analysis, T.W.; investigation, T.W., Z.Z., and F.Y.; resources, Y.Z.; data curation, T.W.; writing—original draft preparation, T.W.; writing—review and editing, Y.Z.; visualization, T.W.; supervision, Y.Z.; project administration, Y.Z.; funding acquisition, Y.Z.

**Funding:** This research was funded by National Natural Science Foundation, grant number 51674022.

**Conflicts of Interest:** The authors declare no conflict of interest.

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
