# Peer review of "Effects of Fe2O3 on Reduction Process of Cr-Containing Solid Waste Self-Reduction Briquette and Relevant Mechanism"

_metals, doi:10.3390/met9010051_

Reviewer 1 Report

Dear Authors

your paper “Effects of Fe2O3 on Reduction Process of  Cr-containing Solid Waste Self-reduction Briquette  and Relevant Mechanism” discusses about the influence of Fe2O3 on the CrOx reduction in self-reducing briquettes made by stainless steel production residues.

The paper is interesting since it aims to find the optimum conditions to recover Cr from specific metallurgical residues, with special regards to the possibility to add iron oxides to increase the reduction yield of Cr.

Even the paper is well structure, in some point it must be revised. Currently, it does not meet the requirement for publication on metals journal

In the following a list of suggestions/corrections/remarks that I would like to pose to the Authors attention.

Abstract

p.1, line 14-15. Please check the sentence because I cannot understand to what is referred the term “thermodynamically”

Introduction

p.1, line 21-22: About the Cr oxidation in stainless production, could the Authors specify at which step of the process they are referring? If the stainless steel are produced by AOD converter, Cr2O3 is recovered by deoxidation of slag and the residual concentration of Cr in the slag is approximately 1-3%. Is this marginal concentration valuable to be recovered in Authors opinion?

Experimental procedure

p.2, Table 1: as the Cr concentration in the stainless steel slag and dust is quite high, may I suppose that such residues came from EAF production step. Could the Authors clarify their source?

p.3, Table 2: what nc means? Number of C moles? Could the Authors specify below no in the last row of the table? Same for R.

p.3, line 76: why MgO crucible? Have the Authors note some reactions between crucible and FeOx/SiO2? At the experimental temperature, MgO is quite reactive with FeOx and SiO2 to form fayalite-forsterite solid solution, which posses low melting temperature. Could this aspect affect the reduction behavior?

Results and discussion

Equation (1): I did not agree with the Authors about the method to define the reduction rate.

First because the equation 1 express a sort of reduction yield or reduction efficiency instead of reduction rate and secondly, because to define the whole reduction yield, also O removal from CrOx and FeOx must be accounted. In my opinion a better coefficient would be the ratio between the actual mass loss and the theoretical mass loss, calculated as the maximum mass loss due to C oxidation and CrOx and FeOx reduction to Cr and Fe.

p. 4, line 107-118. The discussion of Figure 4 should be updated as a function of the correct reduction yield index recalculated as suggested before. About the explanation given, I think that briquette A and B behave at the same way during quenching, since it was performed in inert atmosphere as stated by the Authors. Why briquette B should oxidized during quenching and A-type not? In addition, the Authors mention high-Cr-metal and low-Cr-metal but no results are shown in this term. Please provide results that clearly show the difference in metallic Cr content of the reduced briquettes or modify the discussion accordingly. The same for activity coefficient and free energy discussion. Shows the free energy calculation before discussing its decreasing or increasing

p.4, line 124: I suppose the Authors would mean Reaction 4 instead of reaction 3.

p.5, line 136: How can the Authors state that BL5 sample has violent reaction?

p.5, Figure 8: sorry but I cannot find correspondence between spectrum #13 and elemental maps. By elemental maps, the coarse and edge-shaped phase marked as #13 should not have Cr neither Fe inside, but high concentration of Al and O. Please could you check it?

p. 10, line 212-213: What the Authors would mean with the term “polymerized” referred to a metal?

p.10, line 215-216: about the viscosity, could the Authors refer to a paper or a book? In the present form, this assumption is just a conjecture, not proven by experimental results.

p.11, Figure 13: in a very simplified way, the reducibility of CrOx in comparison with FeOx can be studied directly by Ellingham diagram in standard state. Cr → CrOx equilibrium has an oxygen potential more negative than Fe → FeOx equilibrium. In addition, C → CO equilibrium overcomes  Cr → CrOx only at very high temperatures. Thus, Fe2O3 helps the reduction of CrOx because it can be fastly reduced to a liquid and than, the liquid can dissolve CrOx favoring its subsequent reduction. The discussion proposed by the Authors achieve the same result in a more complicate way.

Conclusions

p. 13, line 279: this conclusion should be updated after reduction yield equation correction

References

reference list is a bit poor. I suggest the Authors to increase the number of records and using it to reinforce their discussion

p { margin-bottom: 6.25px; line-height: 120%; }

Author Response

Dear editors and reviewers:

The authors are great appreciated that the reviewers made important comments regarding the manuscript entitled “Effects of Fe2O3 on Reduction Process of Cr-containing Solid Waste Self-reduction Briquette and Relevant Mechanism” (No.: metals-402179). All the comments are very valuable and helpful for revising and improving our paper, and they also have important guiding significance to our next research works. We have studied comments carefully and have made corresponding revisions recommended by the reviewers. The revised part of the manuscript has been clearly highlighted using the "Track Changes" function in the enclosed document. The detailed revisions in the manuscript and the responses to the reviewer’s comments are as follows.

Reviewer 1

Comment 1

Abstract

p.1, line 14-15. Please check the sentence because I cannot understand to what is referred the term “thermodynamically”

Response (p.1, line 16-18)

Here, thermodynamically was used to express the meaning of “A large number of Fe-C alloy droplets generated in the lower temperature could decrease the activity of reduced chromium by in situ dissolution and the reduction of Cr-oxide was accelerated.”. We have rewrote the sentence.

Comment 2

Introduction

p.1, line 21-22: About the Cr oxidation in stainless production, could the Authors specify at which step of the process they are referring? If the stainless steel are produced by AOD converter, Cr2O3 is recovered by deoxidation of slag and the residual concentration of Cr in the slag is approximately 1-3%. Is this marginal concentration valuable to be recovered in Authors opinion?

Response: (p.2, line 55)

Thanks for your comment on this point. There mainly have two kinds of stainless steel slags, EAF slag and AOD slag. The former usually has higher content Cr2O3 (3-14%) based on our investigation. The latter has lower content Cr2O3 (<3%) because of the high-efficient recovery rate of AOD furnace. In this work, EAF slag with high Cr concentration was chosen for the experiments. However, marginal concentration valuable was not conformed because the possibility of Cr leaching also existed even though the content of Cr2O3 was lower than 3%。So, we thought the economical deep-reduction of CrOx in the slag should be achieved. The marginal concentration may be based on the some national standard.

Comment 3

Experimental procedure

p.2, Table 1: as the Cr concentration in the stainless steel slag and dust is quite high, may I suppose that such residues came from EAF production step. Could the Authors clarify their source?

p.3, Table 2: what nc means? Number of C moles? Could the Authors specify below no in the last row of the table? Same for R.

p.3, line 76: why MgO crucible? Have the Authors note some reactions between crucible and FeOx/SiO2? At the experimental temperature, MgO is quite reactive with FeOx and SiO2 to form fayalite-forsterite solid solution, which posses low melting temperature. Could this aspect affect the reduction behavior?

Response: (p.2, line 55, p.3, line 87)

p.2, Table 1: Yes, the slag and dust samples respectively came from EAF and AOD production steps in an enterprise in north china. And, their source has been clarified.

p.2, Table 1: Yes, the nC means the number of C moles for graphite and nO means the number of O moles in FeOx and Cr2O3 from slag, dust and Fe2O3. And, the means of them has been specified.

p.3, line 76: Thanks for your comments for this point. MgO crucible has good corrosion resistance from metal and slag products. Compared with Al2O3 crucible, it does not crush under Ar-quench. In addition, during 5 minutes, the C in the briquette has reacted with most of FeOx and Cr2O3, thus little amounts of FeOx could react with MgO after slag-metal separation. In addition, SiO2 was used to adjust final slag basicity (R=1.2) and made slag with good fluidity. Some MgO eroded, the SiO2 added, and the CaO, MgO, and Al2O3 from materials could form a favourable slag, which ensure the separation between metal and metal.

Comment 4

Results and discussion

Equation (1): I did not agree with the Authors about the method to define the reduction rate.

First because the equation 1 express a sort of reduction yield or reduction efficiency instead of reduction rate and secondly, because to define the whole reduction yield, also O removal from CrOx and FeOx must be accounted. In my opinion a better coefficient would be the ratio between the actual mass loss and the theoretical mass loss, calculated as the maximum mass loss due to C oxidation and CrOx and FeOx reduction to Cr and Fe.

Response: (p.4, line 103-115)

Yes, yours comments on that is accurate. We have corrected the term “reduction rate” to your suggestion “reduction yield” due to its rationality. And the discussion of Figure 4 has been updated.

p. 4, line 107-118. The discussion of Figure 4 should be updated as a function of the correct reduction yield index recalculated as suggested before. About the explanation given, I think that briquette A and B behave at the same way during quenching, since it was performed in inert atmosphere as stated by the Authors. Why briquette B should oxidized during quenching and A-type not? In addition, the Authors mention high-Cr-metal and low-Cr-metal but no results are shown in this term. Please provide results that clearly show the difference in metallic Cr content of the reduced briquettes or modify the discussion accordingly. The same for activity coefficient and free energy discussion. Shows the free energy calculation before discussing its decreasing or increasing

Response: (p.4, line 131-132)

For the decrease of reduction yield at higher temperature, we though that the B-briquette with higher Cr content (Cr/(Cr+Fe)=0.65) could be easily oxidized at higher temperature due to its higher Cr activity although a quick operation and a high-purity Ar was used. For the low Cr content (Cr/(Cr+Fe)=0.23), the effect would be not obvious.

p.4, line 124: I suppose the Authors would mean Reaction 4 instead of reaction 3.

Response: (p.5, line 141-142)

Yes, it has been corrected.

p.5, line 136: How can the Authors state that BL5 sample has violent reaction?

Response: (p.6, line 157-158)

It can be seen that there are many pores and holes caused by gas product in the BL5 sample, and its final product is higher than other samples. So, we can conclude a violent reaction has happened. But, no clear metal particles are formed, and slag and metal mixes together. Its description has been supplied.

p.5, Figure 8: sorry but I cannot find correspondence between spectrum #13 and elemental maps. By elemental maps, the coarse and edge-shaped phase marked as #13 should not have Cr neither Fe inside, but high concentration of Al and O. Please could you check it?

Response:

We have checked the correspondence between spectrum #13 and elemental maps. The phase 13# mainly contained elements Cr, Fe, Al, Mg, O, and it is a kind of spinel phase originated from materials. The corresponding element distribution has been cycled in the following figure. Please check it.

p. 10, line 212-213: What the Authors would mean with the term “polymerized” referred to a metal?

Response: (p.6, line 234-235)

It means that the metal droplet with low Cr and high Cr content contacted and dissolved each other. We have checked it.

p.10, line 215-216: about the viscosity, could the Authors refer to a paper or a book? In the present form, this assumption is just a conjecture, not proven by experimental results.

Response: (p.6, line 234-235)

The high-melting point M7C3 (Cr7C3: 1780°C) suspended in the metal product usually causes viscous metal which has some solid fraction. Its viscosity can be characterized by the prediction model of solid-liquid coexistence presented by Mori and Ototake [24]. So, we can conclude that the existence of M7C3 increased the viscosity of the metal. The paper has been referred.

[24] Y. Mori and N. Ototake. Viscosity of solid-liquid mixture [J]. Chemical Engineering, 1956, 20: 488.

p.11, Figure 13: in a very simplified way, the reducibility of CrOx in comparison with FeOx can be studied directly by Ellingham diagram in standard state. Cr → CrOx equilibrium has an oxygen potential more negative than Fe → FeOx equilibrium. In addition, C→CO equilibrium overcomes  Cr→CrOx only at very high temperatures. Thus, Fe2O3 helps the reduction of CrOx because it can be fastly reduced to a liquid and than, the liquid can dissolve CrOx favoring its subsequent reduction. The discussion proposed by the Authors achieve the same result in a more complicate way.

Response: (p.12, line 258-268)

Yes, your explanation on this is very concise and accurate. The Figure 13 is the Ellingham diagram in standard state for the Cr or Fe-containing phases in the self-reduction briquette, and it gives detailed initial reduction temperature information in the Cr-containing solid waste self-reduction briquette. We just want to give a detailed presentation and description. We deleted the Equations 7-14 and the Reaction mechanism part seems to be brief.

Comment 5

Conclusions

p. 13, line 279: this conclusion should be updated after reduction yield equation correction

Response: (p.12, line 297-299)

We have corrected the conclusion after updating reduction yield equation.

Comment 6

References

reference list is a bit poor. I suggest the Authors to increase the number of records and using it to reinforce their discussion

According to your suggestion, We have reriched the reference list and reinforced our discussion. The number ofreferences has been enriched from 13 to 27.

Once again, we appreciate for editor and reviewers’ precious comments and suggestions. Hoping the revision will meet with approval. Looking forward to hearing from you.

Yours sincerely,

Tuo Wu

[email protected]

Reviewer 2 Report

The article is interesting and offers information abouts the effects of Fe2O3 in the reduction of Cr-containing briquettes. However, the document should be reviewed according to the comments provided in the enclosed pdf. Additionally, the number of references is too limited, and the English should be reviewed as some sentences are not clear due to this question.

Author Response

Dear editors and reviewers:

The authors are great appreciated that the reviewers made important comments regarding the manuscript entitled “Effects of Fe2O3 on Reduction Process of Cr-containing Solid Waste Self-reduction Briquette and Relevant Mechanism” (No.: metals-402179). All the comments are very valuable and helpful for revising and improving our paper, and they also have important guiding significance to our next research works. We have studied comments carefully and have made corresponding revisions recommended by the reviewers. The revised part of the manuscript has been clearly highlighted using the "Track Changes" function in the enclosed document. The detailed revisions in the manuscript and the responses to the reviewer’s comments are as follows.

Reviewer 2

Page 1 Line 12: . Moreover, the relevant mechanism is also studied.

Response: (p.1, line 12-13)

The sentence has been revised.

Page 1 Line 15: What is "thermodynamically"?. Rewrite the sentence it is not clear what do you mean with the word thermodynamically

Response: (p.1, line 12-13)

Here, thermodynamically was used to express the meaning of “A large number of Fe-C alloy droplets generated in the lower temperature could decrease the activity of reduced chromium by in situ dissolution and the reduction of Cr-oxide was accelerated.” The sentence has been rewritten.

Page 1 Line 34: what do you want to say with incubation period?

Response:

The “incubation period” came from the paper wrote by Görnerup et al.[15-16], which means that the reduction of Cr2O3 by C at certain temperature does not take place until some FeO has been reducted.

[15] Görnerup, M.; Lahiri, A.K. Reduction of electric arc furnace slags in stainless steelmaking Part 1 Observations. Ironmak. Steelmak. 1998, 25, 317-322.

[16]  Görnerup, M.; Lahiri, A.K. Reduction of electric arc furnace slags in stainless steelmaking: Part 2 Mechanism of CrOx reduction. Ironmak. Steelmak. 1998, 25, 382-386.

Page 1 Line 40: It is not clearly indicated why FeOx has influence in the reduction of Cr briquettes.

Response:

The literatures reported by researchers founded that FeOx plays an important role in the reduction process of CrOx. So, for the Cr briquette with much Cr-containing phases, we must pay attention to the FeOx content in the Cr briquette and its relevant mechanism, which were shown in the following part of the report.

Page 1 Line 41: What is RB? reduction briquette?

Response: (p.2, line 46)

Yes, it is the abbreviation of “Cr-containing solid waste self-reduction briquette”.

Page 2 Line 51: Metallic iron only or also magnetite?

Response: (p.2, line 57)

Only metallic iron was removed by a magnet. Because the C content added was based on FeOx in the slag according to nC:nO.

Page 2 Line 65: Is this high melting point problematic for the process? What is the approximate temperature? Working at high temperatures would mean important energy requirements, was evaluated if recovering copper compensates the costs of this high temperature process?

Response: (p.2, line 70)

Yes, Magnesiochromite (MgCr2O4) normally occurs in the stainless steel slag and has high melting point (2390℃) which causes the Cr loss. So, the reduction of Cr in the slag mainly is the reduction of Magnesiochromite. The C or SiFe usually were chosen for the reduction of Cr slag in the plants.

In addition, i don’t understand the meaning of “recovering copper”.

Page 3 Line 75: What is ID? Inner diameter?

Response: (p.3, line 80-81)

Yes, it is Inner diameter.

Page 3 Line 76: What is OD? Outside diameter? What is H? Height? It is necessary to indicate the abbreviations.

Response: (p.3, line 81-82)

Yes, they are Outside diameter and Height.

Page 3 Line 77: BLMT?

Response: (p.3, line 82)

BLMT is one manufacturer for the high-temperature furnace for the high-temperature experiments. The “BLMT-1700” is one kind of furnace version in our lab.

Page 3 Line 79: Around what point?

Response:

The 6 cm is the height of constant temperature zone of furnace tube. In the constant temperature zone, the temperature error was ±2 oC, which does not affect the experiment result in certain temperature (1350, 1450, 1550℃).

Page 3 Line 88: Drawing the zones in the figure 3 might be interesting.

Response: (p.4, line 88)

We have drawn the zones as shown in Figure 3.

Page 3 Line 89: How long was it held in the constant temperature zone?

Response: (p.4, line 94)

As shown in the Table 2, the sample was held in the constant temperature zone for 5, 20, 40 min.

Page 3 Line 93: it and was

Response: (p.4, line 98)

The words have been revised.

Page 4 Line 100: CO is extremely dangerous, did you emitted CO as gas? Moreover, reduction mechanisms usually involve the Boudouard mechanism, and in this way, CO is the reductant of the load (solid-solid reactions are very difficult). You should deeply study this point, and explain the reduction mechanism clearly.

Response:

Because the sample was so small and constant Ar was introduced in the experiment, small amount of CO was diluted and did not cause danger. In our present study, we thought CO may partly play the role of indirect reductant for the Fe2O3 and inverted to CO2, and finally it was the gas product taken away by Ar. The process has been draw in the mechanism diagram.

Page 4 Line 107: on

Response: (p.5, line 121)

The “of” has been be revised as “on” based on your suggestion.

Page 4 Line 109: Please rewrite this sentence, it is not clearly explained: "Figure 4 shows that the reduction rate of the briquettes increases rapidly before 5 min, and then, the increasing rate slows down with the extension of time"

Response: (p.4, line 121-123)

We realized that the “reduction rate” defined in the Equation 1 may cause doubt in this sentence. So we corrected it to “reduction yield”. We have revised the sentence.

Figure 4 shows the time and temperature dependences on reduction rate yield of A and B-type self-reduction briquettes. As shown that the reduction yield of them increases rapidly before 5min and then its increasing rate slows down with the extension of time.

Page 4 Line 119: How can you explain that reduction progresses according to this reaction? Solid-solid reactions are very difficult.

Response: (p.4, line 116)

We thought that the reduction of FeOx and Cr2O3 by C in the briquette according to the reaction equation (4) causes the rapid increase of “reduction yield” in the initial 5min. And in the experiment temperatures such 1350, 1450, 1550℃, the solid-solid reactions in the briquette really happened as shown in Figure 5.

Page 5 Line 131: Rewrite the sentence, it is not clear

Response: (p.6, line 145-147)

We have revised the sentence to the following expression.

It can been seen that as the temperature keeps constant, the metal products in every samples gradually contact each other and grows up, and the metal-slag interface are clearer which means the separation of them are more complete as the reaction time prolongs.

Page 5 Line 137: Where you see a clear separation between slag and metal? What is metal and what is slag?

Response: (p.5, line 141, p.6, line 142)

As shown in Figure 5, the small metal mixed with slag or non-reacted reactant, as the temperature increases and time prolongs, the interface of them has become clear. We revised the figure to show the slag and metal and the interface of them clearly.

Page 7 Line 152: Why the higher temperature increases the reduction rate of Fe and Cr oxides? What happens with the decomposition of the oxides with the temperature?

Response: (p.8, line 166)

For the C reduction reaction of FeOx and Cr2O3 at high temperature, the reaction is often exothermal and the conclusion has been referred to the previous result[21,22]. Therefore, higher temperature increases the reduction rate of Fe and Cr oxides. At high temperature, the reduced Fe, Cr and C could dissolve each other to form Fe-Cr-C alloys and separate with slag formed by other oxides such as CaO, SiO2, Al2O3, MgO, which are difficult to reduce.

[21]Zhang Y L, Liu Y, Wei W J. Products of carbothermic reduction of Fe–Cr–O and Fe–Cr–Ni–O systems[J]. Transactions of Nonferrous Metals Society of China, 2014, 24(4):1210-1219.

[22]Zhang, Yan-ling, Liu, et al. Carbothermal reduction process of the Fe-Cr-O system[J]. International Journal of Minerals Metallurgy & Materials, 2013, 20(10):931-940.

Page 7 Line 167: How can you explain the differences in morphology?

Response:

We thought that when the Cr content is higher certain level, the high melting point M7C3 ((Cr, Fe)7C3) would precipitate in Fe-Cr-C alloys, which caused the irregular shape of metals. And, the following Fe-Cr-C phase diagram also verified that.

Page 9 Line 194: This means that chromium spinel is difficultly reduced? Is the reduced chromium detected from other kind of phases?

Response: (p.10, line 208)

We have revised the sentence to express that “Compared with the B-type RB without Fe2O3, the Cr-containing phase was not easily reduced in a lower temperature and shorter time”.

Page 13 Line 277: How do you propose the separation metal-slag?

Response:

The separation between slag and metal was induced by the difference of their density, viscosity, surface tension et al. And the metal usually has bigger density and higher surface tension than slag, thus the separation has happened.

Once again, we appreciate for editor and reviewers’ precious comments and suggestions. Hoping the revision will meet with approval. Looking forward to hearing from you.

Yours sincerely,

Tuo Wu

[email protected]

Reviewer 3 Report

Corrections:

Table 1. TFe (or Fe), Line 59

Figure 14: reduction cycle (not recuction cycle), Line 275

1. Which chemical equation corresponds to the calculation  represented by Equation (1)

2. Are you sure that the mass loss contains only carbon monoxide?

3.Did you Analyse the formed gas?

4. Can you give Information about reduction rate after 3 min?

5. Did you perform XRD Analysis in order to define the compounds in final products?

6. Did you perform XRD Analysis in order to confirm the FActSAge Analysis?

7. Did you perform thermalgravimmetric Analysis (mass loss in time)  in order to confirm your reduction product?

8.  What is the role of carbon monoxide in your mechanism?

9. In which form is the produced chromium?

10. How to separate the recovered chromium?

11. What it the particle size of spherical Fe-C alloys? (line 283)

12. Is more important effect of an added Fe2O3 or an injection of carbon monoxide?

Author Response

Dear editors and reviewers:

The authors are great appreciated that the reviewers made important comments regarding the manuscript entitled “Effects of Fe2O3 on Reduction Process of Cr-containing Solid Waste Self-reduction Briquette and Relevant Mechanism” (No.: metals-402179). All the comments are very valuable and helpful for revising and improving our paper, and they also have important guiding significance to our next research works. We have studied comments carefully and have made corresponding revisions recommended by the reviewers. The revised part of the manuscript has been clearly highlighted using the "Track Changes" function in the enclosed document. The detailed revisions in the manuscript and the responses to the reviewer’s comments are as follows.

Reviewer 3

Corrections:

Table 1. TFe (or Fe), Line 59

Figure 14: reduction cycle (not recuction cycle), Line 275

Response: (p.2, line 64, p.2, line 13, 287)

Table 1. : It is really the “TFe”, which means “Total Fe”.

Figure 14. : The wrong “recuction” has been revised to “reduction”.

1. Which chemical equation corresponds to the calculation represented by Equation (1)

Response: (p.4, line 116)

The Equation (1) was calculated by the weight loss after the quench of samples, and the theoretical mass loss calculated as the maximum mass loss due to assumed emission of CO produced after the complete reduction of Cr2O3 and FeOx in the self-reduction briquette.

The chemical Equation 4 was corresponded to calculate the maximum mass loss due to assumed emission of CO.

2. Are you sure that the mass loss contains only carbon monoxide?

Response: (p.4, line 108-112)

Since reduction experiments were carried out at relatively high temperatures (≥1350 oC), in which CO was more stable than CO2 and high-purity argon would take the gas product continuously, and an excessive amount of C could react with CO2 produced by indirect reduction (Equation 2) to form CO based on Boudouard reaction (Equation 3), it was assumed that the gas product is mainly CO.

3.Did you Analyse the formed gas?

Response:

It is with regret that we did not analyse the formed gas because of no available equipment. We analysed the conditions the experiments involved, and the excess C and the high temperature could ensure that the gas was mainly formed with CO.

4. Can you give Information about reduction rate after 3 min?

Response:

Since the experiment was operated at a high temperature, the 3min was too short to get a good result and cause large error, especially for the three different temperatures. So the reduction rate after 3min was not acquired, and we thought the result at 5min has shown an relatively objective information, which was the entire reduction time for the Cr-RB.

5. Did you perform XRD Analysis in order to define the compounds in final products?

Response:

We have performed the XRD analysis of AL5 and BL5 samples, and the result has been shown in the following Figure. As shown that the Ca2Mg(Si2O7) and Fe-Cr alloy were founded in the two samples. The difference is that the AL5 and BLB has C and (Fe, Cr)7C3 respectively, which confirmed the FactSAge Analysis.

6. Did you perform XRD Analysis in order to confirm the FActSAge Analysis?

Response:

The XRD analysis was performed and the result conformed the FActSAge Analysis as shown in the above response.

7. Did you perform thermalgravimmetric Analysis (mass loss in time)  in order to confirm your reduction product?

Response:

We understand that “Thermalgravimmetric Analysis” may better reveal the detailed information in the Cr-RB. However, in the present study, we mainly focused on the effects of Fe2O3 on reduction process of Cr-containing solid waste self-reduction briquette, and we think that SEM+EDS and XRD suppled may not be optimal, but should be sufficient to draw a conclusion that the importance of Fe2O3 added in the Cr recovery. In addition, results of TG are unfortunately unavailable in a short time.

8. What is the role of carbon monoxide in your mechanism?

Response:

In our present study, we thought CO may partly play the role of indirect reductant for the Fe2O3 and inverted to CO2, and finally it was the gas product taken away by Ar.

9. In which form is the produced chromium?

Response: The briquette made of stainless steel solid wastes was heated and reduced at high temperature, and Fe-Cr alloy and slag with low concentration Cr was produced. The former could be recycled for steelmaking and the latter could be disposed with no consider chromium leaching.

10. How to separate the recovered chromium?

Response:

This study was conducted to figure out the effect of Fe2O3 on the reduction process made of stainless solid waste and further emphasized the importance of Cr/(Fe+Cr) parameter for the self-briquette. For the industrial production, the recyclable molten Cr-Fe alloys could separate with the slag due their density difference at high temperature.

11. What it the particle size of spherical Fe-C alloys? (line 283)

Response:

Based on the SEM+EDS results and theoretical analysis, we deduced that the Fe-C alloys formed in the initial reduction process and the particle size of spherical Fe-C alloys at 5 min and 1350℃ was counted and calculated. It was about 13-110μm and the average is 45μm.

12. Is more important effect of an added Fe2O3 or an injection of carbon monoxide?

Response:

Cr-oxide usually was reduced by C according to Cr2O3→Cr3C2→Cr7C3→Cr23C6→Cr process. In our research, we found the CrxCy as intermediate products could delay the reduction process. So FeOx was introduced to make a metal bath to dissolve the CrxCy. And C play an important role in the direct reduction and generation of Fe-C-(Cr) bath to accelerate the reaction. For the dust which contained mainly FeO•Cr2O3, CO could not reduce it. So the an added Fe2O3 has more important effect.

Once again, we appreciate for editor and reviewers’ precious comments and suggestions. Hoping the revision will meet with approval. Looking forward to hearing from you.

Yours sincerely,

Tuo Wu

[email protected]

Round  2

Reviewer 1 Report

Dear Authors Thank you for your kind and precise revision. The paper gains quality and it can be a good starting point for industrial scale up. I have some minor revisions to pose to your attention: 

1) figure 1: please add the scale bar to the pictures.

2) figure 4: authors stated that the reduction yield increase rapidly before 5 min treatment. However this can be deduced by the figure only because origin and first point are connected. May the authors check the reduction yield at 2.5 min to be sure that the fast increasing is not an artefact due to the graph?

3) may the authors add some final remarks in conclusion section about the scale-up feasibility of the proposed process (which kind of reactor would be better? Rotary, hearth, fluidized bed...)

Best regards

Author Response

Dear editors and reviewers:

On behalf of my co-authors, we thank you very much for giving us an opportunity to revise our manuscript again. We also appreciate editor and reviewers very much for their positive and constructive comments and suggestions on our manuscript entitled “Effects of Fe2O3 on Reduction Process of Cr-containing Solid Waste Self-reduction Briquette and Relevant Mechanism” (No.: metals-402179). We have carefully studied the comments given by the reviewer and have made corresponding revisions. The revised part of the manuscript has been marked in blue in the enclosed document. The detailed revisions in the manuscript and the responses to the reviewer’s comments are as follows.

Comment 1

1) figure 1: please add the scale bar to the pictures.

Response: ()

We have added the scale bar to the Figure 1, which was shown in Page 2 Line 63.

Comment 2

2) figure 4: authors stated that the reduction yield increase rapidly before 5 min treatment. However this can be deduced by the figure only because origin and first point are connected. May the authors check the reduction yield at 2.5 min to be sure that the fast increasing is not an artefact due to the graph?

Response:

Since the experiment was operated at a high temperature, the 2.5min was too short to get a good result and cause large error, especially for the three different temperatures. So the reduction rate after 2.5min was not acquired, and we thought the result at 5min has shown a relatively objective information, which was the entire reduction time for the Cr-RB.

In addition, the reduction time for the Cr-RB designed by available materials such slag, dust and Fe2O3 was also evaluated based on the previous reports, which also contained the similar results for only three reduction yields which were plotted the φ vs time Figure.

In order to improve the results obtained by this experiment, we are thinking some improvements for our setup and hope to get more detailed reduction information for the Cr-RB.

Thank you very much for your comment and suggestion.

Comment 3

3) may the authors add some final remarks in conclusion section about the scale-up feasibility of the proposed process (which kind of reactor would be better? Rotary, hearth, fluidized bed...)

Response: (Page 13 Line 291-293)

We thought that the self-reduction briquette could be disposed by high temperature reduction methods. The reactors which can be fed self-reduction briquette such as rotary hearth furnace, shaft furnace and EAF furnace et al. under feasible operations were remarked in the conclusions.

Once again, we appreciate for editor and reviewers’ precious comments and suggestions. Hoping the revision will meet with approval. Looking forward to hearing from you.

Yours sincerely,

Tuo Wu

State Key Laboratory of Advanced Metallurgy

University of Science and Technology Beijing

Reviewer 3 Report

This improved Version shall be accepted in this  present form. Thank you for your Improvement and an additional work. 

Author Response

Thanks for your kindly comment work, which improves our manuscript.

Round  3

Reviewer 1 Report

Thank you for your kind reply

I have not more comments

Best regards